# Atypical Pathogens in Adult Community-Acquired Pneumonia and Implications for Empiric Antibiotic Treatment: A Narrative Review

**DOI:** 10.3390/microorganisms10122326

**Published:** 2022-11-24

**Authors:** Nicolas Garin, Christophe Marti, Aicha Skali Lami, Virginie Prendki

**Affiliations:** 1Division of Internal Medicine, Riviera Chablais Hospital, 1847 Rennaz, Switzerland; 2Division of General Internal Medicine, Geneva University Hospital, 1211 Geneva, Switzerland; 3Faculty of Medicine, University of Geneva, 1211 Geneva, Switzerland; 4Division of Infectious Disease, Geneva University Hospital, 1211 Geneva, Switzerland; 5Division of Internal Medicine for the Aged, Geneva University Hospital, 1211 Geneva, Switzerland

**Keywords:** pneumonia, atypical, empiric treatment, *Mycoplasma pneumoniae*, *Legionella pneumophila*, *Legionella longbeachae*, *Chlamydia pneumoniae*, *Chlamydia psittaci*, *Coxiella burnetii*

## Abstract

Atypical pathogens are intracellular bacteria causing community-acquired pneumonia (CAP) in a significant minority of patients. *Legionella* spp., *Chlamydia pneumoniae* and *psittaci*, *Mycoplasma pneumoniae*, and *Coxiella burnetii* are commonly included in this category. *M. pneumoniae* is present in 5–8% of CAP, being the second most frequent pathogen after *Streptococcus pneumoniae*. *Legionella pneumophila* is found in 3–5% of inpatients. *Chlamydia* spp. and *Coxiella burnetii* are present in less than 1% of patients. *Legionella longbeachae* is relatively frequent in New Zealand and Australia and might also be present in other parts of the world. Uncertainty remains on the prevalence of atypical pathogens, due to limitations in diagnostic means and methodological issues in epidemiological studies. Despite differences between CAP caused by typical and atypical pathogens, the clinical presentation alone does not allow accurate discrimination. Hence, antibiotics active against atypical pathogens (macrolides, tetracyclines and fluoroquinolones) should be included in the empiric antibiotic treatment of all patients with severe CAP. For patients with milder disease, evidence is lacking and recommendations differ between guidelines. Use of clinical prediction rules to identify patients most likely to be infected with atypical pathogens, and strategies of narrowing the antibiotic spectrum according to initial microbiologic investigations, should be the focus of future investigations.

## 1. Introduction

Community-acquired pneumonia (CAP), i.e., acute infection of the lung parenchyma acquired outside the hospital, is a frequent disease and has a large impact on morbidity and mortality worldwide. According to the Global Burden of Disease collaboration, there were almost 600 million episodes of pneumonia and other lower respiratory tract infections (LRTI) in 2019 globally, causing 2.5 million deaths [1]. Pneumonia incidence and mortality are highly correlated with socio-economic factors. Children less than 5 years old are disproportionately affected in lower outcome countries [2]. In high-income countries, pneumonia mainly affects older people, the incidence ranging from less than 1% in adults less than 50 years, and increasing exponentially thereafter up to more than 35% in people more than 90 years old [3]. Finally, pneumonia accounted for 80% of infectious diseases mortality in the U.S in 2014 [4].

Pneumonia is a highly heterogeneous disease. Firstly, a large number of bacterial, viral, fungal and parasitic pathogens are able to invade the lung, as a single pathogen or in co-infection, though a more limited number cause the majority of microbiologically-documented pneumonia. However, the causative pathogen is often not identified, due to the difficulty to obtain appropriate samples from the lower respiratory tract and to shortcomings of available diagnostic tools. Even with extensive investigations using both culture- and nucleic acid amplification-based methods, the aetiology is typically established in no more than 50% of episodes, and usually far less frequently [5].

Secondly, severity can range from a mild, self-resolving disease managed in the ambulatory setting to a fulminant infection leading to respiratory failure, septic shock and multiple organ failure. Both host- and pathogen-related factors probably explain this wide severity spectrum [6,7].

The fear of a severe evolution leads to most episodes of pneumonia being treated with antibiotics. The infecting agent is generally unknown at the onset of the disease and remains unidentified in at least half of the cases later, though higher rates of pathogen identification can be achieved with comprehensive bacteriological and molecular testing [8,9,10]. The antibiotic treatment is therefore frequently empiric. This means that the choice of the initial treatment is probabilistic, aiming to cover a large number of pathogens, with a special focus on bacteria known to cause severe infection.

The wish to include a higher number of pathogens in the antibiotic spectrum is in tension with the risk of selecting antibiotic-resistant bacteria, both at the individual and at the population level. Because of their frequent occurrence, CAP and LRTI represent the first cause of antibiotic prescriptions in the hospital (ca. 20% of all prescriptions) and in ambulatory care (ca. 40%) worldwide [11,12]. Hence, national and international recommendations for the empiric antibiotic treatment of CAP may have a profound impact on antibiotic prescribing patterns and selection pressure.

The need to add coverage for so-called atypical pathogens is a widely debated issue in this context [13,14]. In the present review, we examine current knowledge on the concept of atypical pathogens, the epidemiology of main members of this category, and the pro-and con-arguments toward their empiric antibiotic coverage. We summarise recent international guidelines and discuss the possibilities to target atypical antibiotic coverage towards patients most likely to benefit. Because the spectrum of disease and possible pathogens are obviously different in severely immunosuppressed patients (e.g., after solid organ or bone marrow transplantation; or patients with acquired immunodeficiency syndrome, or receiving potent immunosuppressive drugs for auto-immune or auto inflammatory diseases), in paediatric patients, and in patients with nosocomial pneumonia, they will not be included in this review.

## 2. From Atypical Pneumonia to Atypical Pathogens and Atypical Coverage

The concept of «atypical pneumonia» dates back to the beginning of the 20th century and is tightly connected to the description of classic, typical, pneumococcal pneumonia, with which it was contrasted.

The early history of pneumonia is dominated by *Streptococcus pneumoniae* [15]. First described at the end of the 19th century, it was an overwhelming cause of death and morbidity at the beginning of the 20th century, causing ca. 85% of pneumonia [5]. Clinically, it presented as an acute disease beginning with chills, high fever and pleuritic chest pain, progressing in a few hours to days to cyanosis, confusion, respiratory distress and death [16]. Once sulphonamides and penicillin were available, mortality decreased drastically, with a reduction in the absolute risk of mortality between 25 and 65% in controlled studies, depending on the age of the patient and the severity of disease [17].

Besides this «classic» form of pneumonia, other presentations were described during the first half of the 20th century. The term «atypical pneumonia» was used for a lung infection with clinical and radiological characteristics differing from *S. pneumoniae* infection [18] (Table 1). The main characteristics of atypical pneumonia were a more progressive, subacute course; the presence of pronounced constitutional and extra pulmonary symptoms; non-productive cough; and a better prognosis despite a protracted disease [19,20]. There was no leucocytosis on blood count; Gram stain, blood and sputum cultures did not show evidence of *S. pneumoniae* (nor other known bacterial pathogens like Klebsiella sp. or *Staphylococcus aureus*); and chest X-ray did not show lobar infiltrates, but scattered, ill-defined opacities. Finally, penicillin or sulphonamides did not alter the course of the disease. The non-bacterial (at least according to the conceptions at the time) nature of the infecting pathogen causing atypical pneumonia was correctly predicted from demonstrating that it was transmissible to humans by sputum despite a filtration process removing all bacteria (hence the term «non-filtrable» pathogen). After the 2nd World War, the demonstration that the course of atypical pneumonia could be modified by aureomycine (a tetracycline antibiotic) reinforced the need to differentiate typical from atypical pneumonia in order to administrate the correct antibiotic treatment [19].

However, it was already felt that atypical pneumonia was a syndromic presentation due to a set of pathogens differing in their nature and contracted in different epidemiological conditions [19]. Some were endemic (psittacosis, Q fever) [21]; some were clearly epidemic, either during the cold season (influenza) or in young patients living in crowded conditions like military recruits (adenovirus, *Mycoplasma pneumoniae*). Eaton agent, first identified in 1944, later named *Mycoplasma pneumoniae*, was shown to be the cause of epidemics among adolescents and young adults. Though the natural course of the disease was often mild, it could clearly be shortened by tetracyclins [22].

The identification of *Legionella pneumophila* in 1977 was a game changer. Though the clinical presentation differed from pneumococcal pneumonia (frequent gastrointestinal symptoms; dry cough; infrequent leucocytosis), it also differed from commonly observed atypical pneumonia because of the acute beginning, the radiological findings (with more extensive and sometimes unilateral consolidation) and the often severe clinical course with shock, respiratory failure and high mortality (16%) [23]. Since that moment, the use of an antimicrobial treatment active against both *L. pneumophila* and *S. pneumoniae* was warranted in severe pneumonia, thus excluding beta lactam monotherapy. Later, other species of *Legionella* have been identified as pathogens.

Limitations of the syndromic approach (i.e., the ability to predict the implicated pathogen in an individual patient based on differences between atypical and typical presentation) was evident in subsequent works. In a study using a multivariate model prediction, and despite some characteristics being clearly different between the two groups, less than 50% of patients could be correctly attributed to pneumococcal or mycoplasmal pneumonia [24]. Similarly, comparisons of *Legionella* and pneumococcal pneumonia showed substantial overlap in the clinical, biological and radiological presentation [25,26]. The demonstration that the aetiology of pneumonia could not be reliably predicted by the clinic and radiologic presentation led to less emphasis on the syndromic approach and less use of the locution «atypical pneumonia».

However, the term «atypical» survived to describe a special group of pathogens. Though there is no official or universal definition, atypical pathogens commonly include intracellular bacteria not identifiable by standard blood or sputum cultures, and intrinsically resistant to antibiotics inhibiting cell wall synthesis like beta lactams: *Legionella* and *Chlamydia* spp.; *Mycoplasma pneumoniae;* and, less commonly, *Coxiella burnetii*. Prominent authors have strongly criticized the use of this terminology, arguing (appropriately) that *M. pneumoniae* and *Legionella* spp. are quite frequently and consistently identified pathogens and as such do not qualify for the term atypical [27]. However, the category «atypical pathogens» is still largely used in recent publications.

By extension, the concept of «atypical coverage» refers to the empiric use of antibiotics active against atypical pathogens, principally macrolide, tetracycline or fluoroquinolone [28,29,30]. This concept is less clear, as some of these antibiotics are not consistently active against all atypical pathogens (e.g., resistance to macrolide in some strains of *M. pneumoniae*). Moreover, use of «atypical coverage» leads to oversimplification, implying that any benefit with the use of one of these classes, alone or combined with a beta lactam, should be attributed directly to the treatment of atypical pathogens rather than intrinsic properties of the drug (e.g., anti-inflammatory activity of macrolides).

Consequently, some recent highly cited guidelines [31], though not all [32], do not use the term atypical any more, neither for pathogens nor for antibiotic coverage.

Though we recognize the weakness of this classification, the terms atypical pathogens and atypical coverage are still widely in use. We will go on using them in this review with the following definition: an atypical pathogen, a bacterial pathogen that is intracellular or paracellular, not identifiable by gram stain and traditional culture media, and intrinsically resistant to beta lactams; and atypical coverage: antibiotics with efficacy towards intracellular pathogens, mainly macrolides, fluoroquinolones and doxycycline (Table 1).

**Table 1 microorganisms-10-02326-t001:** Atypical pneumonia, atypical pathogens, atypical coverage [18,19,20,27].

	Atypical	Typical
**Pneumonia**		
Clinical course	Subacute onset Protracted disease	Abrupt onset
Symptoms	Extrapulmonary and pulmonary (flu-like illness, myalgias, rhinorrhea, odynophagia, diarrhea, prominent headache) Dry cough; scant sputum	Confined to the lung Pleuretic chest pain Productive cough with coloured sputum
Leucocytosis	Absent	Present
Gram stain, blood and sputum cultures	No evidence of a pathogen	*Streptococcus pneumoniae* (or *Klebsiella pneumoniae*, *Staphylococcus aureus*…)
Chest X-ray	Patchy, ill-defined infiltrates, scattered on both lungs	Lobar pneumonia, pleural effusion
Prognosis	Often favourable, even without antibiotics	Significant mortality despite penicillin
**Pathogens**		
	*Mycoplasma pneumoniae* *Legionella pneumophila* and non-pneumophila *Chlamydia pneumoniae* and *psittaci* *Coxiella burnetii* (*Francisella tularensis*; *Bordetella pertussis*)	*Streptococcus pneumoniae* *Hemophilus influenzae* *Moraxella catarrhalis* *Klebsiella pneumoniae* *Staphylococcus aureus* *Streptococcus* sp. (*Pseudomonas aeruginosa*; other Gram-negative enterobacteriaceae)
**Antibiotic coverage**		
	Macrolides Tetracyclines Fluoroquinolones	Betalactams Aminoglycosides Respiratory Fluoroquinolones (Macrolides and Tetracyclines)

## 3. Epidemiology

Although they have been described for decades, uncertainties remain on the incidence of pneumonia caused by atypical pathogens. Reports vary widely, ranging from less than 5% to more than 20% of patients affected by CAP [33,34]. A striking example is *Legionella* infection, which is a reportable disease in most countries, allowing geographical comparisons. Incidence in 2012 in high-income countries ranged from 0.02/100,000 inhabitants (Poland) to 4.02/100,000 (Slovenia), a ratio of 200 in reported incidence without a sound biological explanation, highlighting differences in identification and reporting of the disease [35]. A widely cited study conducted on 4337 patients from 21 countries, using oropharyngeal swabs for culture and PCR, paired blood samples for serology, and urine for *Legionella* antigen detection found a global incidence of 22%, with few variations between continents [33]. In another monocentric cohort from Spain, including 3523 patients recruited during 12 years and with extensive testing, atypical pathogens were identified in 263 (18% of patients with an identified pathogen, 7.5% of the total population) [6].

In contrast, at least one test to search for any atypical pathogen was done in only 34% of patients worldwide (46% in Europe) in the GLIMP database (3702 patients). The vast majority of tests performed was urine testing for Legionella antigen. The prevalence of atypical pathogens was only 4.7% [34]. A study combining four European cohorts for a total of 3297 patients with various testing strategies for atypical pathogens found a prevalence of 3% (14% of patients with a pathogen identified) [36].

Finally, a meta-analysis conducted in 2016 included 30 studies reporting the prevalence of *M. pneumoniae*, *L. pneumophila* and *C. pneumoniae* [35]. Studies had to include consecutive adults independently of the clinical setting (ambulatory or hospital). Prevalence was 7.2% (range < 1% to 24%) for *M. pneumoniae*, 2.8% (range 1% to 10%) for *L. pneumophila*, and 4.3% (range < 1% to 21%) for *C. pneumoniae*. Heterogeneity was very high (I^2^ > 90% for all three pathogens) [35]. In a recent systematic review, studies were stratified by the microbiological methods used (standard cultures only; cultures plus serology; and cultures or serology plus PCR [9]). *M. pneumoniae* represented 8.9–10.5%, *Chalmydophila* 3.1–5.3%, and *Legionella* 6.2–6.6% of patients in which an aetiologic agent was identified. As the corresponding proportions are not reported for the whole population (i.e., all patients with pneumonia, including those without an identified pathogens), these figures are not directly comparable with the 2016 meta-analysis.

The explanations for this heterogeneity are multiple. Sampling during an epidemic is a possible source of variation, especially concerning *M. pneumoniae*, which exhibits cyclic epidemics recurring approximatively every 4 years [37,38]. Seasonal variations also affect the relative incidence of some atypical bacteria. Unlike most respiratory pathogens, *Legionella* spp. is more frequently found during summer and fall [39]. The clinical context is also important: *M. pneumoniae* and *C. pneumoniae* cause predominantly mild disease, rarely needing hospitalization, and will be more frequent in studies including patients with less severe disease [6,40]. Other variations can also arise from differences in the investigations performed: more efforts to obtain lower respiratory tract samples with fibroscopy or induced sputum lead to a higher diagnostic yield [41,42]. This could have special relevance for older patients, for whom good quality expectorations can be difficult to obtain, and who are also less prone to undergo invasive procedures [43].

Finally, major differences can stem from the microbiologic test used to make the diagnosis. Atypical pathogens are fastidious to grow in culture and diagnosis traditionally relied on acute and convalescent serum samples or demonstration of IgM in acute serum [44,45]. Since the introduction of polymerase chain reaction (PCR) in routine clinical practice, PCR on sputum or upper respiratory swabs (oro- or nasopharyngeal) tends to substitute for serology or culture. PCR has a lot of advantages over cultures, including ability to detect fastidious organisms, identification of viral pathogens, timeliness of the results, better sensitivity, and ability to detect bacteria after antibiotic administration [46,47]. Drawbacks include lack of standardization and difficulties in differentiating colonization from infection [48].

However, concordance between PCR, serology and culture is far from perfect, different PCR have various accuracies, and oro- and nasopharyngeal swabs have less sensitivity and specificity than lower respiratory tract samples [49,50]. Additionally, asymptomatic carriage can lead to false-positive diagnosis of *M. pneumoniae* pneumonia [51].

Serologic studies constantly report higher incidence of atypical infections than culture or PCR studies. Specificity of serology is only moderate for *M. pneumoniae* [52]. This lack of specificity is also true for *C. pneumoniae*: though prevalence in CAP can be as high as 20% in serologic studies, it is detected in less than 1% of cases in PCR-based studies [53,54,55]. In the absence of a reference diagnosis, it can be difficult to assess if under- or over diagnosis is present in an individual study. Finally, co-infections (the identification of more than one pathogen) with a typical bacteria or a virus are described in 14–25% of cases of documented atypical pathogen infection in studies using comprehensive microbiological testing [6,56,57].

The additional yield of comprehensive testing (i.e., using both culture- and nucleic acid amplification) is illustrated in a study conducted by Gadsby et al. [8]. In 323 adults able to produce good quality sputum, a pathogen (either viral or bacterial) could be identified in 87% of patients by using a combination of multiplex PCR assays compared with 39% with culture-based methods. The atypical pathogens detected were 6 *Legionella* spp., 3 *C. psittaci*, and 3 *M. pneumoniae*. Another study suggested a decrease from 65 to 43% in the proportion of patients without an identified pathogen [10].

A balanced evaluation for adult patients hospitalized for pneumonia is that *M. pneumoniae* is probably present in 5–8% of patients (second bacterial pathogen in frequency after *S. pneumoniae*) and *Legionella* spp. in 3–5% [35,58]. Major uncertainties remain on the true prevalence of *C. pneumoniae*.

## 4. *Legionella* spp.

The bacteria *Legionella* was first identified as a human pathogen in the investigation of an outbreak affecting a meeting of the American legion in 1976 [23,59]. The genus *Legionella* consists of ca. 90 species [60]. The most known species of the genus is *L. pneumophila*, which can be subdivided further in serogroups. *L. pneumophila* caused the initial outbreak in 1976 and is implicated in the majority of human infections reported, though this could reflect failure to search for other species of *Legionella* [61,62]. Indeed, more than 20 species of *Legionella* are able to infect humans. *Legionella* spp. mostly affects the lower respiratory tract. Self-limited forms of the disease present as a flu-like syndrome without pneumonia and are named «Pontiac fever». Lung infection caused by *Legionella* spp. is called legionellosis. Clinical and radiologic presentation of pneumonia is similar between all *Legionella* species [63]. Finally, *Legionella* spp. can rarely cause non-respiratory infections: skin and soft tissue infections, arthritis, endocarditis and meningo-encephalitis [61,63]. Risk factors for legionellosis are older age, smoking, chronic lung disease, and immunosuppression, including chronic glucocorticoid treatment. Unlike most respiratory pathogens, legionellosis predominates in summer and fall [39,64].

*Legionella* are Gram-negative bacteria living as intracellular parasites of amoebae or as free-living bacteria in biofilms [60]. This co-evolution with amoebae probably explains *Legionella*’s outstanding adaptation to eukaryotic cells and their ability to harness metabolic and signalling host cell pathways towards its own survival and growth, and to inhibit normal cellular defence mechanisms [60,61,63].

*Legionella* gains access to lung alveoli via aerosols or micro aspiration. Intact cellular immunity is probably paramount to protect against infection. *Legionella* is phagocytosed by alveolar macrophages. It is then able to inhibit the fusion of the phagosome with the lysosome and to alter the phagosome to transform it in its niche, called LCV (*Legionella*-containing vacuole), where it is able to grow and replicate. Finally, apoptosis of the host cell is triggered, and the bacteria is released and enters a new cycle of infection [60,61,63].

*Legionella pneumophila* is found ubiquitously in natural aquatic environments (lakes and rivers) and artificial water reservoirs and pipes (cooling towers, air-conditioning systems, showers…) [60,63]. Though most cases are sporadic, outbreaks can occur from contaminated sources like cooling towers, water systems, including, rarely, hospital equipment [39]. The incubation period is about 7 days.

*Legionella longbeachae* is identified in most non-pneumophila legionellosis. It is the predominant species of *Legionella* in New Zealand and is also frequently identified in Australia [61,64]. Unlike *L. pneumophila, L. longbeachae* is primarily found in potting spoils and compost, and epidemiologic evidence links *L. longbeachae* infection with gardening [61,65,66,67].

*L. pneumophila* causes 80–90% of reported cases of legionellosis in Europe and North-America, and serogroup 1 is responsible for 90% of these cases [63]. However, diagnostic bias could lead to underestimation of the incidence of non-pneumophila *Legionella* infection, as the most used diagnostic tool, urinary antigen detection, only detects the *L. pneumophila* serogroup 1 efficiently [68]. *L. pneumophila* was causing only 20–25% of legionellosis in a national surveillance study conducted in New Zealand when sputum of patients hospitalized with pneumonia was routinely tested by PCR for the presence of *Legionella* spp. [64]. The vast majority of other legionellosis were caused by *L. longbeachae*. An additional finding was that the incidence was three-times higher than the previous years, hence pointing to under detection of legionellosis. Descriptions of clusters of *L. longbeachae* infections in Scotland and Sweden suggest that this pathogen should be considered as a cause of pneumonia also in Europe [8,69,70].

*Legionella* spp. (and particularly *L. pneumophila*) infections are reportedly increasing worldwide. It is unknown if this corresponds to a real increase in incidence, secondary to climatic change, more susceptible people (through ageing and immunosuppression), or artefactual, secondary to more awareness and more testing [64,71].

The clinical presentation of Legionellosis differs somehow from typical pneumonia by a longer prodromal illness, a higher fever, more extra-pulmonary symptoms (particularly gastro-intestinal) and neurologic findings (acute confusion) [25,63]. However, as discussed previously, the clinical picture is not discriminant enough to allow an accurate diagnosis of *L. pneumophila* based on clinical findings.

It was initially thought that *L. pneumophila* caused mostly a severe disease, as it was overrepresented in studies conducted in intensive care units [72]. However, more recent studies have shown that *L. pneumophila* is also the cause of milder severity pneumonia [6,73]. Mortality of *L. pneumophila* pneumonia probably does not differ from all cause pneumonia [6].

Diagnosis of *L. pneumophila* infection frequently relies on detection of urinary antigen, which is widely available, easy to realize, and relatively low-cost [58]. Specificity of the various antigen tests is excellent, but sensitivity is at best 74% and is restricted to *L. pneumophila* serogroup 1 [74]. Culture of sputum remains the reference standard and allows the detection of all *Legionella* species. However, it is slow (up to seven days), labour-intensive, and requires the inoculation on buffered charcoal-yeast extract medium; as this medium is not routinely used, adequate management of samples requires previous suspicion and laboratory notification. PCR is more rapid than culture, and has good specificity and sensitivity when performed on lower respiratory tract samples [75]; sensitivity is poor on nasopharyngeal swabs [76]. PCR can target *L. pneumophila*, *Legionella* spp., and duplex PCR targeting both *L. pneumophila* and *L. longbeachae* are available [77]. Paired serum sampling is the method of choice for epidemiologic studies; however, serology is not useful in clinical practice. Seroconversion can take as long as two months, sensitivity is limited, and high titers can persist several years after an infection [78].

Because of its intracellular growth, *Legionella* spp. is resistant to beta lactam drugs and aminoglycosides. Assessing antibiotic activity against *Legionella* is difficult with conventional methods because the agar used binds the antibiotics [63]. Both macrolides and fluoroquinolones achieve high intracellular concentrations and have good activity in-vitro; levofloxacin and azithromycin are considered as the reference treatment. A recent meta-analysis found no difference in efficacy between fluoroquinolones and macrolides [79]. Doxycycline is considered active against *L. pneumophila*, but *L. longbeachae* might be resistant, and doxycycline should not be used when *L. longbeachae* infection is suspected [80].

## 5. *Mycoplasma pneumoniae*

Mycoplasma are among the smallest free-living organisms. Among 120 species, only four are well-known as human pathogens, including *Mycoplasma pneumoniae*, which causes principally respiratory tract infections and can rarely have manifestations outside of the respiratory system, mostly immune-mediated [81,82].

*M. pneumoniae* lacks a cell wall; consequently, it is not visible on Gram stain and is intrinsically resistant to antibiotics inhibiting cell wall synthesis, like beta-lactams [82]. Inter human transmission occurs via droplets and causes epidemics in persons in close contact with a cumulative attack rate as high as 90 percent. In England, recurrent epidemic periods occur at 4-yearly intervals [83]. Average incubation lasts 2 to 3 weeks. Recently, incidence of *M. pneumoniae* declined worldwide after implementation of non-pharmaceutical interventions against COVID-19 [84]. Asymptomatic carriage is frequent as well as prolonged carriage after symptomatic infection, with a median duration of approximately seven weeks, which plays an important role in its transmission [51,85].

*M. pneumoniae* is the most commonly identified atypical pathogen, especially in mild to moderate pneumonia [86,87,88]. *M. pneumoniae* causes upper respiratory infections and acute bronchitis and is a common bacterial cause of CAP. It was found in 2 to 12 percent of adults hospitalized for CAP in an US prospective cohort and 6.8% of patients included in the large German CAP-Competence Network [86,87,88]. In children aged ≥5 years and hospitalized with CAP, it was the most frequently detected bacteria, along with another pathogen in one-quarter of the cases [89].

Headache, malaise, low-grade fever, sore throat, cough, pleuritic chest pain and shortness of breath are frequently observed [81,82,90]. *M. pneumoniae* infection may worsen asthma symptoms and produce wheezing [91]. The course is generally mild, even without antibiotics. Fulminant cases with respiratory failure and death are exceptionally reported [92].

Mild haemolysis or elevation of hepatic enzymes are present in half of the patients and are rarely symptomatic [81,93]. Haemolysis can be occasionally severe in patients with underlying hematologic disorders such as sickle cell disease. Haemolysis is immune mediated, driven by induced cold agglutinins targeting antigens on red blood cells. Among other less frequent manifestations, *M. pneumoniae* can affect the central nervous system and cause encephalitis, meningitis, transverse myelitis, Guillain-Barré syndrome, cranial nerve palsies, and cerebellar ataxia [94]. Other systems can be affected, probably also by immune-mediated mechanisms, including the heart (pericarditis, myocarditis, cardiac thrombi, and conduction abnormalities), the skin (erythematous maculopapular or vesicular rashes, urticaria, erythema multiform, Stevens–Johnson syndrome), and the musculoskeletal system (arthralgia and myalgia) [81,90,93].

The radiographic features of *M. pneumoniae* pneumonia are similar to other atypical or viral pneumonias. A chest radiograph may reveal reticulonodular or patchy unilateral or bilateral opacities [90]. One study performing systematic high-resolution CT-scan described frequent lateral bronchial wall thickening coupled with minimal air bronchograms [95]. However, these findings are not specific enough to distinguish *M. pneumoniae* pneumonia from other interstitial pneumonias.

Multiplex PCR-based assays can be performed on respiratory tract samples (e.g., nasopharyngeal swab, sputum, bronchoalveolar lavage fluid) [82,96]. However, PCR cannot distinguish between active infection and asymptomatic carriage. The Biofire FilmArray respiratory panel which includes *M. pneumoniae* has been approved by the FDA for the diagnosis of respiratory tract infections [97]. Cultures are fastidious and generally not used for the diagnosis. Direct Coombs test and cold-agglutinin titres are typically positive in the presence of haemolysis [98].

Macrolides, doxycycline and fluoroquinolones are all active against *M. pneumoniae* [81]. No direct comparison has been made in a randomized-controlled trial. However, an observational study using propensity-score matching in 1650 Japanese patients did not find any significant difference in efficacy between these three antibiotic categories; no test for resistance to macrolides was available in this study [99]. Macrolide or doxycycline are generally proposed as first line therapy [100,101].

Very rare before 2000, macrolide resistance first emerged in Japan and the Far-East and has steadily increased among *M. pneumoniae.* Macrolide-resistant *M. pneumoniae* are now present worldwide [102,103]. In a recent systematic review, the prevalence of macrolide resistance was 53% in the Western Pacific region, 10% in the South East Asian region, 8% in the Americas, and 5% in Europa [104]. Macrolide resistance is associated with prolonged symptoms in patients treated with macrolides, but not with a higher rate of complications [105]. Treatment with tetracyclines or fluoroquinolones seems effective and is the recommended option when a patient infected by *M. pneumoniae* fails to improve on macrolide treatment [100,103,106].

## 6. *Chlamydia pneumoniae*

*C. pneumoniae* is a very small obligate intracellular bacteria which belongs to the *Chlamydiaceae* family and *Chlamydia* genus [107]. *Chlamydia* and *Chlamydia*-related bacteria may be the agents of pneumonia of unknown aetiology. The cell wall of *Chlamydia* spp. has an inner and outer membrane, but its peptidoglycan is present in small quantities, which implies a natural resistance to beta lactams. Existing as a small, dense elementary body (EB) when outside the host, *C. pneumoniae* becomes a metabolically active reticulate body (RB) after entering respiratory mucosal epithelial cells. After replication, EBs are released and infect new cells. According to in vitro data, *Chlamydiae* could cause persistent infection and play a role in chronic illnesses [107].

The prevalence of *C. pneumoniae* infection varies according to studies and depends on the clinical presentation, the timing (epidemic and clusters) and diagnostic methods used, ranging from 1 to 20% of cases of CAP (with a higher prevalence in mild cases) [6,108]. Studies using PCR have found a much lower prevalence [53,109]. Prevalence was 0.9% in adults included in the German CAP Competence Network [55].

*C. pneumoniae* is transmitted between humans via droplets, aerosols, and fomites. Outbreaks have been reported in people living in close quarters, with an attack rate of 34% [110,111].

Pneumonia caused by *C. pneumoniae* has a nonspecific presentation and is usually mild, with fever, cough, and shortness of breath [112]. The majority of infections are asymptomatic and severe cases are exceptionally described. Non-respiratory manifestations are rare and include meningoencephalitis, Guillain-Barré syndrome, myocarditis and endocarditis [107].

Chest radiograph findings are nonspecific. Microbiological testing is usually indicated in case of severe pneumonia. PCR-based testing can be performed on nasopharyngeal swabs, sputum, and bronchoalveolar lavage fluid [113]. Many multiplex PCR respiratory panels include *C. pneumoniae* and have been approved by FDA [114]. Serology is rarely used and is not useful for diagnosis of CAP [55].

## 7. *Chlamydia psittaci* and Psittacosis

Psittacosis is a zoonotic infection. *Chlamydia psittaci* is an obligate intracellular organism transmitted to humans from birds, which are the primary reservoir. At least 460 species have been described in many bird orders, from pet to poultry and wild birds [115].

Epidemiology is unprecise because of lack of testing and varying performance of diagnostic tests. Psittacosis is found in ca. 1 percent of CAP [116]. Most patients have a history of contact with birds in domestic settings or at work, but sometimes this exposition may lack [53]. Infection in birds is usually asymptomatic. *C. psittaci* is shed in faeces, urine, and respiratory secretions. Humans are usually infected by inhalation of dry organisms which remain viable in dried faeces for months. Cases of psittacosis acquired from cats and dogs have been described [117]. Human-to-human transmission may rarely occur [118].

*C. psittaci* incubation usually lasts from 5 to 14 days, with an attack rate of 10% [119]. Infection can be asymptomatic or lead to mild disease. Fever, rigors, myalgia, headache—which may be severe—dry cough, pharyngitis, diarrhoea, delirium and hepatosplenomegaly can be observed [120]. Severe pneumonia and respiratory failures are rare. Neurologic, renal, gastrointestinal, cardiac, haematological and liver complications may rarely occur and can be serious. Infection in pregnancy can be life-threatening, especially in the second or third trimester; foetal outcome is poor.

The chest radiograph is usually abnormal and most often shows lobar changes. C-reactive protein and procalcitonin are elevated [121]. Culture is difficult and hazardous. Serology is the principal diagnostic method and micro immunofluorescent antibody test is preferred to complement fixation.

Polymerase chain reaction (PCR) methods have been developed for the detection of *C. psittaci* but are not yet commercialised. A study using multiplex PCR assays to assess patients with pneumonia found that the prevalence of psittacosis was higher than estimated [122].

## 8. *Coxiella burnetii*

Q fever is a worldwide zoonosis caused by *Coxiella burnetii*, identified after an outbreak in abattoir workers in Queensland, Australia, in 1935 [123]. *C. burnetii* may cause a wide spectrum of both acute and chronic manifestations. *C. burnetii* is a strict intracellular bacterium, usually hosted by macrophages. An antigenic shift helps differentiating acute from chronic Q fever [124]. When expressing phase I antigen, *C. burnetii* is highly infectious, unlike the phase II form. Many animals, including ticks can be reservoirs. Farm animals are the most commonly identified source of human infection. Infected mammals shed *C. burnetii* in their urine, faeces, milk and placenta. Contamination may occur through inhalation of aerosols and persons at risk are farmers, veterinarians, or abattoir workers [125]. Infection can also occur via transplacental transmission, intradermal inoculation, blood transfusion, or consumption of raw milk. Infection during pregnancy can lead to spontaneous abortion, premature labour, intrauterine growth retardation and intrauterine death. Human to human transmission is rare [123].

Q fever incubation lasts about 20 days. The most common manifestations are fever, fatigue, headache, and myalgias. Fever can persist three weeks. In case of pneumonia, patients present with cough and fever, but respiratory failure is rare. Extrapulmonary manifestations include severe headaches, myalgias, arthralgias, rash and pericarditis, myocarditis and aseptic meningitis or encephalitis. In French Guiana, CAP represents 90% of *C. burnetii* infections and prevalence of *C. burnetii* among CAP is 38.5% [126].

Chest radiograph and laboratory findings are unspecific findings. Antiphospholipid antibodies and lupus anticoagulant may be found.

Acute Q fever is usually diagnosed if the anti-phase II is ≥200 for IgG and ≥50 for IgM, or if anti-phase II IgG is fourfold increased by immunofluorescence assay on serum taken three to six weeks apart [127]. Seroconversion is usually detected one to two weeks after the onset of clinical symptoms. PCR testing can be performed on blood or tissue samples in patients with a clinical suspicion and for whom the initial serologic testing reveals no or low levels of antibodies. It remains positive for 7 to 10 days in acute infection.

## 9. Evidence Regarding Empiric Coverage of Atypical Pathogens

As discussed above, the atypical pathogen probably accounts for a minority of CAP, with important variations according to the setting. Moreover, clinical presentation and usual diagnostic tests lack specificity and sensitivity and often fail to identify a specific pathogen as the cause of CAP [43]. Finally, the results of these etiologic investigations are rarely available immediately. For these reasons, initial antibiotic therapy in CAP remains mainly empirical [128]. The adjunction of “atypical” coverage has been a matter of debate, and this appellation may be misleading [129,130,131]. Indeed, benefits of combination of antibiotics (beta-lactams and macrolides for example) might encompass effective treatment of atypical pathogens, synergistic or adjunctive effect on typical pathogens or intrinsic properties of the drug (e.g., anti-inflammatory activity of macrolides) [27]. 

The benefits of atypical pathogen coverage were evaluated in several observational and randomised controlled trials (RCTs) [29,132,133,134,135].

In a 2012 systematic review, Eliakim-Raz et al. identified twenty-eight RCTs comparing antibiotic regimens with or without atypical coverage [134]. The atypical antibiotic was administered as monotherapy in all but three studies and only one study compared a beta lactam therapy combined with a macrolide to the same beta lactam. Overall, no mortality reduction was observed in the atypical coverage arm (RR 1.14; 95%CI 0.84 to 1.55). Similarly, no mortality reduction was observed in the subgroup of studies (*n* = 19) using quinolones as atypical coverage (RR0.98; 95%CI 0.69 to 1.39). However, this evidence was limited by the use of heterogeneous antibiotic regimens, often differing from current recommended treatments.

The benefit of macrolide adjunction was further investigated in a Swiss RCT comparing beta-lactam monotherapy to macrolide-beta-lactam combination (BICAP study) [132]. This multicentre RCT including 580 immunocompetent non severe inpatients failed to demonstrate the non-inferiority of beta-lactam monotherapy. Absolute difference in clinical stability at day seven was 7.6% (95%CI −0.8 to 16%), in favour of the combination arm. In the 31 (5.3%) patients with an atypical pathogen, the HR for clinical stability was 0.33 (95%CI 0.13 to 0.85), contrasting with an HR of 0.99 (95%CI 0.80 to 1.22) in the subgroup of patients without identified atypical pathogens.

In a subsequent cluster-randomised trial, Postma et al. compared sequential antibiotic strategies including beta-lactam monotherapy, beta-lactam/macrolides combination and respiratory fluoroquinolones [133]. The crude 90-day mortality was 9%, 11.1% and 8.8%, respectively, during these strategy periods. Considering a 3% absolute risk difference in 90-day mortality as the upper non-inferiority limit, the authors concluded with the non-inferiority of beta-lactam monotherapy compared to the combination or fluoroquinolones’ strategies. However, these contrasting results deserve some comments.

First, the non-inferiority boundary used in the CAP-START study (absolute 3% risk difference) appears high, as it corresponds to a relative increase of about one third in the risk of early mortality considering a 9% baseline mortality. Second, deviations from the randomly allocated antibiotic strategy were frequent in the CAP-START study. Thirty-nine percent of patients allocated to the beta-lactam arm received atypical coverage during hospitalisation which may have biased the results towards the null hypothesis. Third, atypical pathogens were identified in only 2.1% of patients in the CAP-START study, which is lower than in the BiCAP study (5.3%). Finally, these contrasting results must be considered in the context of observational studies, suggesting a relative reduction in short term mortality using beta-lactam/macrolide combination or fluoroquinolones compared to beta-lactam monotherapy [131].

Taken together, the available evidence suggests a possible, albeit small reduction of adverse outcomes in favour of atypical coverage in CAP. The mechanisms of the risk reduction with atypical pathogen coverage remain unclear as well as the optimal duration of combination therapy. Although available studies compared combination therapy during full treatment duration, current clinical practice consisting of interrupting atypical coverage after negative testing (usually only with *Legionella* urinary antigen) for atypical pathogens should be evaluated in future RCTs.

## 10. An Overview of International and National Guidelines on Empiric Antibiotic Treatment for CAP

Many national and international boards have issued recommendations of empiric antibiotic treatment for CAP. Some are based on critical appraisal of the literature after systematic research, though the amount of available evidence is often limited [31,32]. Others reflect mainly expert opinion. Guidelines generally acknowledge the importance of local epidemiological data to tailor recommendations.

When deciding on the preferred treatment option, guidelines have to balance patient-centred and population-centred considerations. The best interest of the individual patient is to receive a treatment active against the most likely pathogens, especially those known to cause severe pneumonia. The latter include *S. pneumoniae*, *S. aureus*, *Legionella* spp., and Gram-negative bacteria like *Haemophilus influenzae* and *Klebsiella pneumoniae*. Conversely, *M. pneumoniae*, *C. burnetii* and *Chlamydia* spp. are generally considered to cause mostly mild disease with low associated mortality. Lack of severe or frequent side effects, oral biodisponibility and costs are other patient-centred considerations.

On the population level, selection pressure exerted by the various antibiotic classes used on pathogens and on the microbiome are major considerations [136,137]. Of special concern is the potential for selection of multidrug resistant pathogens (both Gram-positive and Gram-negative), and of *Clostridioides difficile*. Costs are another consideration, especially in low-income countries.

Some recent national and international guidelines are summarized in Table 2.

All guidelines use the site of care as a proxy to stratify the severity of CAP (mild severity: ambulatory; moderate severity: non-intensive care unit inpatient; severe: intensive care or intermediate-care unit). A broader antibiotic spectrum is recommended with increasing severity. All guidelines recommend coverage of atypical pathogens for patients with severe CAP, based on the risk of *Legionella* infection. Empiric atypical coverage in mild or moderate severity of disease is more divergent; it is often proposed on an individual basis (“if atypical infection is suspected”) [32]. However, atypical infection being unpredictable on a clinical basis, this recommendation is somehow useless. Japanese guidelines differ in proposing a clinical prediction rule to identify patients with an atypical pathogen [100,138]. This rule has been reported to have moderate accuracy and has, to our knowledge, never been validated outside of Japan, where legionellosis may be less frequently found [140]. Japanese guidelines are also unique in proposing empiric treatment of macrolide-resistant *M. pneumoniae* with fluoroquinolones if warranted by local epidemiologic data [100]. Amoxicillin and doxycycline, alone or in combination, are frequently proposed in mild pneumonia. Macrolides are rarely proposed alone, reflecting the growing resistance of *S. pneumoniae*. Finally, cephalosporin and fluoroquinolones are frequently reserved as a second choice, due to their propensity to select for multidrug resistant pathogens and *C. difficile* infections [141].

## 11. Clinical Prediction Models

Though the aetiology of pneumonia cannot be reliably predicted on a clinical basis, some differences are consistently described between patients infected by typical and atypical pathogens. Clinical prediction rules have been built in an effort to help in the prediction of CAP aetiology, thus allowing for targeting empiric atypical coverage towards patients most likely to benefit.

Some of the prediction models are restricted to the identification of legionellosis, with the assumption that other atypical pathogens are less likely to adversely affect the prognosis if not treated by the initial antibiotic treatment. The quality of derivation and validation studies is variable, some deriving and testing the accuracy of a score on a convenience sample of patients with known pathogens (e.g., comparing a cohort of patients with legionellosis with a cohort of patients with *S. pneumoniae*), which may artificially inflate their accuracy and obviously does not reflect the target population on which the prediction model should apply [142,143,144]. The variables included in the different prediction models are presented in Table 3.

The Japanese Respiratory Society proposed a rule to streamline atypical coverage in CAP with mild or moderate severity. Five clinical and one laboratory variable are used, with one point assigned for the presence of each item. A cut-off of four or more points was reported to have 77% sensitivity and 93% specificity to predict the presence of an atypical pathogen [138]. However, the original study referenced in the guideline is only available in Japanese. In a subsequent study, sensitivity was 85% for *M. pneumoniae*, but only 18% for *L. pneumophila* [142]. The rule has not been tested in other geographical settings. The Community-Based Pneumonia Incidence Study group (CBPIS) proposed a prediction rule for *L. pneumophila* based on four clinical and three laboratory variables, with a weighted score. Sensitivity was 32–51% and specificity 86–95% at the higher cut-off of ten points in two independent validation studies [142,143]. Using a lower cut-off increased the sensitivity to 89–96% but specificity fell to 17–35%. The score proposed by Fiumefreddo et al. is based on two clinical and four laboratory variables [145]. The presence of two or more items had 78% sensitivity and 79% specificity for the presence of *Legionella*. Validation in independent studies found a sensitivity of 94–97% and a specificity of 23–49% [142,147,148]. Recently, Chauffard et al. proposed a simple score aiming to rule-out the presence of atypical pathogens based on four clinical and one laboratory variable [146]. At a cut-off of <2 points, the score had a sensitivity of 100% and a specificity of 35%, allowing in theory to safely withdraw atypical coverage in 33% of patients. However, this score has not been externally validated.

Other prediction rules have been proposed but are either not externally validated [149], or use a large number of different variables [144]. Finally, no clinical rule as yet has been tested in an impact study, a necessary step before large scale use could be considered.

## 12. Conclusions

Atypical pathogens are intracellular bacteria naturally resistant to beta lactam drugs and share some common differences with classic pneumococcal pneumonia. However, the clinical and radiological presentation cannot discriminate atypical pathogens from other bacteria causing CAP, and clinical prediction rules either have insufficient accuracy or have not been adequately validated. Microbiological investigations have several drawbacks, including inadequate sensitivity or specificity, technical difficulties or lack of standardization. Consequently, a lot remains unknown regarding the incidence of CAP caused by atypical pathogens, resulting in heterogeneity between international guidelines and uncertainty on the best empiric antibiotic strategy for patients with mild to moderately severe CAP. Consensus exists with regard to severe CAP, and empiric treatment of atypical pathogens should be used for these patients, even when the initial diagnostic workup is negative for *L. pneumophila*.

Large scale studies should be conducted to investigate the incidence of atypical pathogens, notably *Legionella* other than pneumophila, with adequate sample size and diagnostic means. Current clinical practice consisting of interrupting atypical coverage after negative testing for atypical pathogens should be evaluated in future RCTs.

## Figures and Tables

**Table 2 microorganisms-10-02326-t002:** Comparison of some international guidelines with regard to atypical coverage. Recommendations specific to atypical coverage are in bold characters.

**(a) Mild CAP, ambulatory patients**				
**ATS/IDSA (2019)** [31]	**NICE (2019)** [32]	**South Australian Guidelines (2021)** [101]	**Japanese Respiratory Society (2016)** [100,138]	**ERS/ESCMID (2011)** [139]
No comorbidities Amoxicillin or Doxycycline or Macrolide ^1^ With co-morbidities Amoxicillin/clavulanate or 2nd Cephalosporin AND Macrolide or Doxycycline OR Respiratory Fluoroquinolones	Amoxicillin If penicillin allergy: Doxycycline or Clarithromycin	Amoxicillin AND/OR Doxycycline ^2^ If penicillin allergy Cefuroxime AND/OR Doxycycline If penicillin and Cephalosporin allergy: Doxycycline	Penicillin +/− beta-lactamase If atypical pathogens suspected ^3^ Macrolides OR Tetracycline (Fluoroquinolone) ^4^	Amoxicillin or Tetracycline If penicillin allergy: Tetracycline or Macrolide ^1^ If high bacterial resistance rates against all first-choice agents: Levofloxacin or Moxifloxacin
^1^ if local pneumococcal resistance < 25% ^2^ Initial monotherapy with doxycycline if atypical pathogens suspected based on epidemiology or the clinical presentation ^3^ According to the Japanese scoring system ^4^ A fluoroquinolone should be used if there is high local prevalence of macrolide-resistant *M. pneumoniae*				
**(b) Moderate severity CAP, inpatients, not-admitted to the intensive care unit**				
**ATS/IDSA (2019)** [31]	**NICE (2019)** [32]	**South Australian Guidelines (2021)** [101]	**Japanese Respiratory Society (2016)** [100,138]	**ERS/ESCMID (2011)** [139]
Beta-lactam AND Macrolide OR Monotherapy with respiratory Fluoroquinolone ^1^ OR Beta-lactam AND Doxycycline ^2^	Amoxicillin if penicillin allergy: see Table 2a If atypical pathogens are suspected WITH Clarithromycin	Benzylpenicillin AND Azithromycin if penicillin allergy Ceftriaxone AND Azithromycin if penicillin and Cephalosporin allergy: Moxifloxacin	Penicillin (+/− beta-lactamase) OR Cephalosporin OR Carbapenem Atypical pathogens suspected ^3^ Tetraycline OR Macrolide (Fluoroquinolone) ^4^	Aminopenicillin ± Macrolide OR Aminopenicillin/beta-lactamase ± Macrolide OR Non-antipseudomonal Cephalosporin OR Cefotaxime or Ceftriaxone ± Macrolide OR Levofloxacin OR Moxifloxacin OR Penicillin G ± Macrolide Regular coverage of atypical pathogens may not be necessary in non-severe hospitalized patients.
^1^ Levofloxacin, Moxifloxacin, or Gemifloxacin ^2^ for adults with CAP who have contraindication to both macrolides and fluoroquinolones ^3^ According to the Japanese scoring system ^4^ A fluoroquinolone should be used if there is high local prevalence of macrolide-resistant *M. pneumoniae*				
**(c) Severe CAP, admitted to the intensive care unit**				
**ATS/IDSA (2019)** [31]	**NICE (2019)** [32]	**South Australian Guidelines (2021)** [101]	**Japanese Respiratory Society (2016)** [100,138]	**ERS/ESCMID (2011)** [139]
Beta-lactam AND Macrolide OR Beta-lactam AND respiratory Fluoroquinolone ^1^	Amoxicillin/clavulanate AND Clarithromycin if penicillin allergy: Levofloxacin	Ceftriaxone AND Azithromycin if penicillin and Cephalosporin allergy: Moxifloxacin	No co-morbidities: Fluoroquinolone or Macrolide ^2^ AND Penicillin (+/− beta-lactamase) With co-morbidities: Carbapenem AND Fluoroquinolones or Macrolide or Tetracycline OR 3rd or 4th generation Cephalosporin + Clindamycin +Tetracycline or Macrolide If allergy to b-lactams: Clindamycin or Vancomycin AND Aminoglycoside + AND Fluoroquinolone	Cephalosporin 3rd AND Macrolide OR Moxifloxacin or Levofloxacin ± Cephalosporin 3rd
^1^ Levofloxacin, Moxifloxacin, or gemifloxacin ^2^ A fluoroquinolone should be used if there is high local prevalence of macrolide-resistant *M. pneumoniae*				

**Table 3 microorganisms-10-02326-t003:** Clinical prediction rules for legionellosis or atypical pathogen.

	CBPIS [143]	JRS [138]	Fiumefreddo [145]	Chauffard [146]
Prediction	*Legionella* spp.	Atypical pathogen	*Legionella* spp.	Atypical pathogen
Age		<60 years		<75 years
Smoking	present			
Co-morbidities		no or mild		heart failure
Season				Fall
Cough		paroxysmal, non-productive	non-productive	
Headache	present			
Vomiting	present			
Chest pain				present
Chest examination		normal		
Fever	increased weight with higher temperature		>39.4 °C	
Leucocytes		<10 G/L		
Creatinine	>88 umol/L			
Lactate dehydrogenase	increased weight with higher LDH		>225 UI/mL	
Sodium			<133 mmol/L	<135 mmol/L
C-reactive protein			>187 mg/L	
Platelets			<171 G/L	

## Data Availability

Not applicable.

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
