# Peer review of "Atypical Pathogens in Adult Community-Acquired Pneumonia and Implications for Empiric Antibiotic Treatment: A Narrative Review"

_microorganisms, 2022, doi:10.3390/microorganisms10122326_

Round 1
Reviewer 1 Report
Thank you for the opportunity to review the manuscript entitled “Atypical pathogens in adult community-acquired pneumonia prevalence and implications for empiric antibiotic treatment: a narrative review.” The review is interesting and well-written, but I have some concerns leading to suggest a major review of the manuscript before a potential acceptance. Please find my comments and concerns below.
Major concerns
- I miss many references in the manuscript; a lot of the paragraphs can’t be the authors unique ideas. If the whole paragraph uses the same references, these should occur not only after the last, but also after the first sentence in the paragraph. Many occasions in the manuscript, e.g., lines 94-104, 223-232, 243-247, 271, 309-312, 313-316, 322-324, 329-331, 333-336, 339-343, 344-346 – and many other more. Please review.
- Line 50: not completely true, or at least dependent on which methods that were used, see for example Shoar et al. 2020 https://doi.org/10.1186/s41479-020-00074-3 and Gadsby et al. 2016 https://doi.org/10.1093/cid/civ1214 and Fally et al. 2021 http://orcid.org/0000-0002-1339-2918
Minor comments and concerns
- Title: consider deleting "prevalence".
- Table 1: please insert references for this overview in the legend or the table.
- Line 112: the pathogen is Chlamydia psittaci, if the authors refer to psittacosis as disease, then this term should not be italic.
- Line 140 and 141: Legionella and Chlamydia should be italic. This also applies many other occasions for other pathogens in the manuscript. Please review the whole manuscript and follow the official species taxonomy nomenclature.
- Lines 162 ff.: consider including Shoar et al. 2020 https://doi.org/10.1186/s41479-020-00074-3, a SR about pneumonia aetiology.
- Lines 176 ff.: consider Gadsby et al. 2016 https://doi.org/10.1093/cid/civ1214 and Fally et al. 2021 http://orcid.org/0000-0002-1339-2918 when elaborating on testing strategies. In these studies, an extensive testing strategy was applied.
- Line 259: reported cases of? Legionellosis, I assume, but please add this information.
- Line 298: probably titres?
- Part 5 of your manuscript: you added information about treatment options in Legionella spp., but not for M. pneumoniae – why?
- General suggestion: consider dividing paragraphs 4, 5, 6, 7, 8 further, e.g., pathogen, epidemiology, symptoms, detection, treatment. Consider moving epidemiology to the paragraphs about the different pathogens and maybe just some few words about inconsistency in prevalence/incidence in paragraph 3.
Best wishes!
Reviewer 2 Report
You reported that review of each pathogens, epidemiology, and article in some guidelines for atypical pathogens in adult community-acquired pneumonia. This theme is thought to be very interesting, but it is need to more detail description and information as described below.
1. You described about Macrolide-resistant M. pneumoniae (MRMP), but you wrote only in 3 lines. Certainly, we are suffered from MRMP in all regions. However, there are high rate of MRMP especially in Asia, there are guidelines of treatment for MRMP. Therefore, you have to add the description of epidemiology and guideline for MRMP.
2. JRS guideline was showed in Table.2, but you wrote the one in 2006. The new one was published in 2017 as described in the article, doi: 10.1016/j.jiac.2021.09.007.
You have to re-check the new guidelines.
3. As you know, due to COVID-19 pandemic, the new technology for diagnosis infections represented by PCR. These diagnose methods are very effective for the exam for atypical pathogens. Therefore, the more detail description of recent diagnostic methods is needed.
Author Response
You reported that review of each pathogens, epidemiology, and article in some guidelines for atypical pathogens in adult community-acquired pneumonia. This theme is thought to be very interesting, but it is need to more detail description and information as described below.
Thank you for your supporting comment and for suggesting to better describe relevant topics . All modifications are highlighted in yellow in the manuscript.
- You described about Macrolide-resistant M. pneumoniae (MRMP), but you wrote only in 3 lines. Certainly, we are suffered from MRMP in all regions. However, there are high rate of MRMP especially in Asia, there are guidelines of treatment for MRMP. Therefore, you have to add the description of epidemiology and guideline for MRMP.
Thank you for your suggestion. We expanded the paragraph on MRMP which now reads :
« Very rare before 2000, macrolide resistance has first emerged in Japan and the Far-East and has steadily increased among M.pneumoniae. Macrolide-resistant M.pneumoniae are now present worldwide.91,92 In a recent systematic review, prevalence of macrolide resistance was 53% in the Western Pacific region, 10% in the South East Asian region, 8% in the Americas, and 5% in Europa.93 Macrolide resistance is associated with prolonged symptoms in patients treated with macrolides, but not with a higher rate of complications. 94 Treatment with tetracyclines or fluoroquinolones seems effective and is the recommended option when a patient infected by M.pneumoniae fails to improve on macrolide treatment. »
We also added the following sentence in the section « An overview of international and national guidelines »
« Japanese guidelines are also unique in proposing empiric treatment of macrolide-resistant M.pneumoniae with fluoroquinolones if warranted by local epidemiologic data. »
- JRS guideline was showed in Table.2, but you wrote the one in 2006. The new one was published in 2017 as described in the article,doi: 10.1016/j.jiac.2021.09.007. You have to re-check the new guidelines.
Thank you. We have updated the references, and modified the Table 2 a, b and c accordingly.
- As you know, due to COVID-19 pandemic, the new technology for diagnosis infections represented by PCR. These diagnose methods are very effective for the exam for atypical pathogens. Therefore, the more detail description of recent diagnostic methods is needed.
We have added the following paragraph in the epidemiologiy section, along with appropriate references :
«PCR has a lot of advantages over cultures, including ability to detect fastidious organisms, identification of viral pathogens, timeliness of the results, better sensitivity, and ability to detect bacteria after antibiotic administration.43,44 Drawbacks include lack of standardization and difficulties in differentiating colonization from infection.45»
Round 2
Reviewer 1 Report
The authors responded appropriately to my queries and the manuscript is improved. No further comments.
Reviewer 2 Report
Dear Author.
Thank you for your revision.
You revised as I indicated. Therefore, your article is appropriate for this journal.